# Optimized Airborne Millimeter-Wave InSAR for Complex Mountain Terrain Mapping

**DOI:** 10.3390/s25020424

**Published:** 2025-01-13

**Authors:** Futai Xie, Wei Wang, Xiaopeng Sun, Si Xie, Lideng Wei

**Affiliations:** 1Beijing Institute of Radio Measurement, Beijing 100854, China; xieft@radi.ac.cn (F.X.); weilideng@tsinghua.org.cn (L.W.); 2State Key Laboratory of Rail Transit Engineering Informatization (FSDI), Xi’an 710043, China; xiesi1829@outlook.com; 3Sichuan Highway Planning, Survey, Design and Research Institute Ltd., Chengdu 610000, China; sunxp05@foxmail.com

**Keywords:** complex mountain terrains, terrain mapping, airborne InSAR, sparse synchronous control

## Abstract

The efficient acquisition and processing of large-scale terrain data has always been a focal point in the field of photogrammetry. Particularly in complex mountainous regions characterized by clouds, terrain, and airspace environments, the window for data collection is extremely limited. This paper investigates the use of airborne millimeter-wave InSAR systems for efficient terrain mapping under such challenging conditions. The system’s potential for technical application is significant due to its minimal influence from cloud cover and its ability to acquire data in all-weather and all-day conditions. Focusing on the key factors in airborne InSAR data acquisition, this study explores advanced route planning and ground control measurement techniques. Leveraging radar observation geometry and global SRTM DEM data, we simulate layover and shadow effects to formulate an optimal flight path design. Additionally, the study examines methods to reduce synchronous ground control points in mountainous areas, thereby enhancing the rapid acquisition of terrain data. The results demonstrate that this approach not only significantly reduces field work and aviation costs but also ensures the accuracy of the mountain surface data generated by airborne millimeter-wave InSAR, offering substantial practical application value by reducing field work and aviation costs while maintaining data accuracy.

## 1. Introduction

Large-scale topographic data, a crucial component of fundamental geographic information, significantly impacts various sectors, including engineering construction, geological disaster prevention, and ecological protection. Efficiently acquiring and processing such data has always been a hot topic in photogrammetric research. With the advancement of satellite technology, techniques for rapidly acquiring digital topographic data over vast areas have gradually matured. For instance, NASA has utilized InSAR technology to collect and disseminate the 30 m resolution Shuttle Radar Topography Mission (SRTM) digital elevation model (DEM), which has been extensively applied in numerous projects [1]. Nevertheless, large-scale topographic map data, especially digital topographic data with a scale finer than 1:5000, continue to be predominantly acquired through aerial photogrammetry. Complex mountainous areas with large undulations, diverse landforms, and variable climate often pose significant challenges to the aerial photogrammetric work. Moreover, airspace restrictions are also a critical factor affecting the use of aerial vehicles for photogrammetric operations. Some areas near airports and cities have congested airspace, making it very difficult to obtain permission for aerial flight operations. The window of opportunity for aerial photogrammetry is rare when both weather and airspace conditions are favorable, thus necessitating technology that is less affected by weather conditions to accurately seize the fleeting opportunity for aerial photogrammetric operations and ensure the acquisition of digital topographic data in complex and cloudy mountainous areas.

Interferometric Synthetic Aperture Radar (InSAR) has emerged as a vital technical means of acquiring high-precision digital topographic data due to its all-weather, all-day capabilities, and strong penetration [2], and has been widely applied in fields such as coal mine deformation, ground subsidence simulation, and modeling [3,4,5,6]. An airborne dual-antenna millimeter-wave InSAR is equipped with two antennas, one for emitting electromagnetic waves and the other for receiving ground echo signals simultaneously. It is unaffected by correlation loss, boasting high resolution, real-time acquisition, and flexible mobility. Furthermore, the system operates with millimeter-band electromagnetic waves, which have short wavelengths and weak penetration, making it more conducive to the acquisition of topographic data [7,8].

Since 1996, Sandia Laboratory in the United States has developed several airborne millimeter-wave systems, such as the Twin-Otter InSAR system, with an elevation accuracy of 0.5 m [9]. In 2011, the Institute of Electronics of the Chinese Academy of Sciences developed a prototype of the airborne millimeter-wave three-baseline InSAR, and the flight test results demonstrated an elevation accuracy better than 1 m [10]. In 2013, the 23rd Institute of China Aerospace Science and Industry Corporation also successfully developed an airborne multi-baseline millimeter InSAR system [11]. These InSAR systems have conducted topographic mapping in many locations [12,13].

However, in actual mapping projects, there are several limiting factors when employing airborne InSAR systems for digital topographic mapping in mountainous areas. The side-looking observation of SAR is prone to forming large areas of shadow and layover, leading to data anomalies in the subsequent topographic data inversion process. On the other hand, to ensure the accuracy of data acquisition, it is necessary to establish ground control points synchronized with the flight, which is very difficult in mountainous areas. The temporarily set up corner reflectors, used as ground control points, need to be recovered, resulting in a significant consumption of manpower costs. In addition, some corner reflectors may be destroyed or moved. Therefore, to ensure the number of effective control points, more corner reflectors need to be deployed in the field, which further increases the workload of field work.

To optimize the issues encountered in practical work, this study aims to address the aforementioned problems by utilizing existing DEM data, simulating shadow and layover based on radar observation geometry, estimating the best flight angle, and designing the flight route according to the shape of the observation area and the overall topography in order to minimize the impact of terrain changes, improve data collection quality, and reduce flight costs. At the same time, ground synchronous control points will be set up in easy areas for the main control points, and control points obtained after the flight for supplementary measurement will be used as auxiliary control points for elevation inversion, which can greatly reduce the requirements for synchronous corner reflector control measurement. This not only greatly reduces the difficulty of field work but also has important significance for improving the safety of the operation and saving the overall cost of data acquisition.

## 2. Flight Path Design

### 2.1. Simulation of Shadow, Overlay, and View Extend

Crafting a flight path plan for airborne InSAR involves a multitude of considerations, including airspace restrictions, terrain impacts, and flight costs. In regions with intricate terrain [14,15], the pronounced undulations of mountainous landscapes can cause the side-looking radar observation mechanism to produce shadows and overlays. These phenomena are primary culprits in preventing the successful inversion of elevation data, ultimately leading to the generation of invalid data areas.

In areas with minimal elevation change, the radar imaging area forms a rectangular region parallel to the flight path. However, in mountainous regions, this imaging area is disrupted by terrain variations and object obstructions, leading to gaps between the observation strips. As depicted in Figure 1, the SAR signal is unable to penetrate the back side of the mountain, which results in strip gaps. Consequently, flight paths in mountainous regions need to be more densely arranged to fill these gaps, thereby increasing aviation operating costs. Moreover, the shadows and overlays caused by terrain variations necessitate even greater overlap in coverage.

Therefore, by combining the local terrain characteristics of the survey area for flight path planning and determining the main flight deflection angle, we can reasonably reduce the impact of terrain on data acquisition quality, providing a reference for the final flight path design.

#### 2.1.1. Simulation Method

Based on the principles of radar observation geometry, the shadow area refers to regions that are inaccessible to the radar beam due to obstructions posed by the terrain. The overlay area denotes regions where the range direction fails to resolve the spatial positional relationships of ground objects, as depicted in Figure 2.

By leveraging the radar observation geometry and digital elevation model (DEM) data, such as utilizing SRTM DEM data, the shadow and overlay can be simulated. The target perspective function *α_m_* is defined, which represents the angle between the line connecting the object (*m*, *h_m_*) to the radar and the vertical direction [15]. The expression is as follows:(1)$$\alpha_{m}=arctan(\frac{x_{m}}{H-x_{m}})$$
where *H* represents the absolute flight height and *x_m_* is the ground distance between the object and the sensor below. The condition for any object (*m*, *h_m_*) on the same azimuth to be in shadow is as follows:(2)$$\theta -\theta_{\mathrm{widh}}/2\leq \alpha_{m}\leq max\left\{\alpha_{1},\cdots ,\alpha_{m-1}\right\}\leq \theta +\theta_{\mathrm{with}}/2$$
where $\theta_{\mathrm{width}}$ represents the radar beam width, $\theta -\theta_{\mathrm{width}}/2$ represents the start of radar beam coverage, and $\theta +\theta_{\mathrm{width}}/2$ represents the end of radar beam coverage.

Assuming the objects (*m*, *h_m_*) and (*n*, *h_n_*) are on the same azimuth, the basis for determining that objects (*m*, *h_m_*) and (*n*, *h_n_*) are overlaying each other is as follows:(3)$$R_{n}\leq R_{m}<R_{n+1}\mathrm{or}\;R_{n}\geq R_{m}>R_{n+1}\;s.t.\left\{\left(m,h_{m}\right),\left(n,h_{n}\right),\left(n+1,h_{n+1}\right)\right\}\notin Shadow\;Area\;m\neq n,m\neq n+1$$
where *R* represents the slant range between the radar and the object.

SRTM DEM data with a grid size of 30 m can be obtained online. Using a Python3.10 with GDAL3.8.4 environment, the DEM data can be loaded as a two-dimensional array. Therefore, based on the above criteria, the overlap, shadow, and view extend can be calculated.

In actual calculations, if the calculation adopts the north-up coordinate system, which uses true north as the *y*-axis and east as the *x*-axis, there is a certain difficulty in traversing grid data. Therefore, DEM data can be rotated according to the flight heading angle and thus aligned to the radar coordinate system, which is shown in Figure 3. This simplifies the calculation process. After the calculation is completed, the results are rotated back to true north and then geo-coded to complete subsequent analysis.

#### 2.1.2. Calculation Method of Area Affected by Terrain

In practical applications, employing the simulation method for calculating shadows and overlays, and traversing all possible headings and strips, the associated data processing demands remain significantly high. Considering the high complexity of precise observation geometry calculations, a raster-based operation method is utilized to first determine the general range of the optimal heading, which can quickly obtain the relationship between the proportion of terrain-affected areas and the heading angle. The specific method is as follows:

Using the R_Index Map (*R_i_m_*) [16] to quickly estimate the suitable heading angle is a per-pixel representation that shows the relationship between the radar acquisition geometry (slant range) and terrain (slope angle and aspect). All terrain effects that may affect or allow the detection area are related to this geometric dependency. The calculation method of R_Index is as follows:
(4)$$\begin{array}{cc} R_{i\_m}= & -sin(({Slope}_{-}map\left[rad\right]\times sin({Aspect}_{-}map\left[rad\right]+\\ & LOSAzimuthangle\left[rad\right])-SARderivedLocalincidenceangle\left[rad\right])) \end{array}$$

Using R_Index, a raster calculator can be employed with conventional GIS software to obtain the slope and aspect of the terrain from DEM data, as shown in Figure 4. The *R_i_m_* can easily be calculated by the raster Band Math Tools of Remote Sensing software or by GDAL tool box.

The target areas affected by terrain are divided into the three levels and represented with corresponding colors, as shown in the Table 1. Figure 5 displays the variation in areas severely affected by terrain under different heading angles.

### 2.2. Flight Path Planning

The design of the flight path primarily involves planning the heading, flight altitude, and swath overlap indicators. By estimating the shadow and overlay, one can gain an overview of terrain influence with heading changes. In actual flight path planning, it is not only necessary to consider the impact of terrain but also, more importantly, to design the flight route according to the shape of the survey area itself, as shown in Figure 6a, where aircraft fly along the long side of the route to reduce the flight time loss caused by the aircraft’s turns and entries into the survey area.

As a rule of thumb, turning time may account for 30% of the effective data collection time. Therefore, the design of the flight path needs to fully simulate the coverage of the flight strip and, as much as possible, reduce the number of flight paths and unnecessary aircraft turns, while ensuring the flight strip covers the entire target area without gaps. The theoretical width of the flight strip can be calculated as follows:*W_strip_* = *H_fly_* × (*tan*(*θ* + *θ_width_*/2) − *tan*(*θ* − *θ_width_*/2))(5)
where the in-area flight altitude *H_fly_* = *max* (*TerrainHeighe* + *safeFilghHeight*).

Referring to the empirical calculation formula for the turning radius of a general civil aircraft, the theoretical turning radius of the aircraft can be calculated and the turning radius is only for reference. When actually executed, the entry order into the flight strip is determined according to the swath width. As shown in Figure 6b, if the flight strips are dense, the execution order of the flight can be adjusted.*R* = *TAS*^2^/(*g* × *tanβ*) (6)

*TAS* is the aircraft’s ground speed, *g* is the acceleration due to gravity, *β* is the bank angle during the turn, and *R* is the turning radius.

### 2.3. Overall Flight Path Design Process

The work flow chart of flight path design is illustrated in Figure 7. This process leverages DEM data from the target area, calculates the corresponding slope and aspect data using GIS tools, and subsequently, in conjunction with radar equipment parameters, computes the R_Index values for various flight deflection angles. This process reveals the relationship between the proportion of the area most affected by terrain and the flight deflection angle. Within the range of flight deflection angles that are less affected by terrain, the flight path is designed according to the shape of the target area, trying to align the flight path with the long side of the target area to reduce flight turns. Finally, the coverage range of the flight path, the distribution of overlay and shadow, and the simulation are performed, and the final flight plan is formulated after estimating the flight cost.

## 3. Reducing the Number of Control Points for Rapid InSAR Mapping

Utilizing InSAR data to determine elevation necessitates correlating phase information with actual elevations [17]. Within an ideal vehicle coordinate system [18], the geometric model for InSAR processing is depicted in Figure 8. Assuming the position of antenna 1 is $A_{1}={\left[x_{a},y_{a},z_{a}\right]}^{T}$ and antenna 2 is $A_{2}={\left[x_{b},y_{b},z_{b}\right]}^{T}$, the interferometric baseline $B=\sqrt{b_{x}^{2}+b_{y}^{2}+b_{z}^{2},}$ $b_{x}=x_{b}-x_{a},\;b_{y}=y_{b}-y_{a},\;b_{z}=z_{b}-z_{a}$. Using antenna 1 as a reference, when $A_{1}={\left[0,0,0\right]}^{T},{\;b}_{x}=x_{b},\;b_{y}=y_{b}$ and $b_{z}=z_{b}$.

From Figure 8, it is known that the expression for *T* is as follows:(7)$$T=[Rcos\left(\theta_{sq}\right),\sqrt{R^{2}{sin}^{2}\left(\theta_{sq}\right)-{\left(H-h\right)}^{2}},H-h{]}^{T}$$

The distance from antenna 2 to the target object is as follows:(8)$${\left|A_{2}T\right|}_{2}=R^{2}+B^{2}-2b_{x}Rcos\left(\theta_{sq}\right)-2b_{y}\sqrt{\left(R^{2}sin\left(\theta_{sq}\right)-{\left(H-h\right)}^{2}\right)}-2b_{z}\left(H-h\right)$$
where $\theta_{sq}$ is the squint angle of antenna 1.

Assuming the distance between antenna 1 and antenna 2 is *R*, the distance difference between antenna 1 and antenna 2 to the target is as follows:(9)$$\Delta R\approx b_{x}cos\left(\theta_{sq}\right)+b_{y}\sqrt{{sin}^{2}\left(\theta_{sq}\right)-{cos}^{2}\left(\theta \right)}+b_{z}cos\left(\theta \right)$$

Combining the above, we can obtain the following:(10)$$h=H-R\left(\frac{-b_{z}\left[b_{x}cos\left(\theta_{sq}\right)-\Delta R\right]}{b_{y}^{2}+b_{z}^{2}}\pm \frac{b_{y}\sqrt{\left(b_{y}^{2}+b_{z}^{2}\right){sin}^{2}\left(\theta_{sq}\right)-{\left[b_{x}cos\left(\theta_{sq}\right)-\Delta R\right]}^{2}}}{b_{y}^{2}+b_{z}^{2}}\right)$$(11)$$\Delta \phi =-\frac{2\pi }{\lambda }\Delta$$(12)$$B_{⊥}=\sqrt{b_{y}^{2}+b_{z}^{2}},b_{z}=B_{⊥}sin\left(\alpha \right),b_{y}=B_{⊥}cos\left(\alpha \right)$$

Based on the InSAR geometric model, the key parameters influencing the target elevation *h* encompass the vertical baseline length (*B*_⊥_), the inclination of the vertical baseline (*α*), the along-track baseline (*B_x_*), the interferometric phase difference (∆ϕ), and the slant range (*R*). Traditionally, when the radar platform flies over the calibration field, the phase can be calibrated using ground control points to obtain the above error parameters that affect elevation accuracy. However, the interferometric calibration parameters will change between different missions and different flight paths, so in practical applications, each flight path needs to be calibrated separately with ground control points to improve overall accuracy [19,20].

Radar control points are typically established by measuring the coordinates of ground control points, encompassing both planar and elevation data. These points are often marked using artificial corner reflectors, with the vertex coordinates of these reflectors representing their three-dimensional spatial coordinates, as illustrated in Figure 9a. Radar corner reflectors are employed to augment the reflectivity of radar signals, thereby enhancing the detection and positioning accuracy of targets. The most prevalent design is the three-sided corner reflector, consisting of three mutually perpendicular planes that form a right-angle triangular prism, with each plane intersecting at 90 degrees, as depicted in Figure 9b. This configuration effectively redirects radar waves back to their source. Control points should be uniformly distributed across the target observation area and must be precisely maintained throughout the aerial photography data collection mission to prevent damage by human interference. The standard practice is to manually measure the placement position of the corner reflector and then install markers, such as nails, to denote these points.

To reduce the situation of artificial corner reflectors being damaged, before the flight plan, in the flight execution survey area, place the corner reflector according to the layout requirements; the vertex of the corner reflector should coincide with the center of the cross of the survey nail. When setting up, the bottom of the corner reflector needs to be leveled and the reflective surface of the corner reflector should be clean, without dust, rain, snow, and other coverings. According to the SAR observation geometry, the orientation of the reflector should be perpendicular to the flight route, so that the signal characteristics of the reflector on the radar image are very significant, as shown in Figure 9b, where two reflectors were placed back to back for two flight paths. Thus, the SAR image’s geographical coordinates can be ascertained, as depicted in Figure 10a, where the corner reflectors stand out with a significant signal against a dark background. This contrast facilitates the establishment of correlations among the image coordinates, actual elevation measurements, and phase value, thereby enabling the precise calibration of the elevation inversion parameters for the specific flight path.

However, in the actual operation process, due to the need to set up the corner reflector in the complex mountain area and to set up before the execution of the flight plan, the time left for the setup is very limited, which will affect its quality; this then becomes the main task that takes time and manpower in the entire flight work. In addition, the corner reflector may be damaged by people, causing the reflector to deviate from the original position and lose its accuracy, as shown in Figure 10b. In this case, more corner reflectors need to be laid out to eliminate the wrong values and ensure the overall quantity of the corner reflectors, which further increases the difficulty and uncontrollability of the field work.

Therefore, this paper introduces an innovative approach that integrates synchronous control with post-flight control for the deployment of control points, significantly alleviating the ground work burden associated with synchronous observations. At present, GPS/IMU technology, which is widely used in aerial photogrammetry [21], can obtain the planar coordinates of the image more accurately by compensating for the motion of the observation sensor through the ground-based station post-differential method. After the flight is completed, areas with less ground obstruction and no sudden changes in elevation for post-height measurement target are selected, thereby supplementing the post-elevation accuracy. Figure 11 shows the supplementary control point measurements after the flight.

In areas with small changes in elevation, the planar measurement error of the control points has a limited impact on the reflection accuracy of the elevation. We artificially added errors to the existing control points, as shown in Figure 12, comparing with the impact on the elevation inversion result experiment, as shown in Table 2. As shown in Table 2, the planar measurement does not cause height inversion error.

Currently, the planar accuracy of aerial photogrammetry assisted by GPS/IMU can reach a centimeter level. Therefore, supplementing the measurement of ground control points after the flight can meet the accuracy requirements for acquiring large-scale topographic data.

Figure 13 delineates the comprehensive process of airborne InSAR, which leverages both synchronous ground control points and post-flight control measurements. Initially, the master and slave strips of SLC data undergo pre-processing steps, including registration, before being subjected to interference processing to extract the original phase information. This phase information is then meticulously calibrated using both the synchronous and post-measured control points. Following this, the phase unwrapping operation and elevation inversion are performed, culminating in the derivation of ground digital model data post-geographic encoding and DSM editing. This meticulous sequence of operations ensures the precision and reliability of the topographic data acquired.

## 4. Results

An area within the southwestern mountainous region of Sichuan was designated as the experimental target for data acquisition. The parameters of the millimeter-wave InSAR radar are detailed in Table 3.

The experimental area consists of mountainous and plain regions as shown in Figure 14, with the mountainous area occupying a majority. Overall, the terrain gradually decreases from an average altitude of 3955 m in the west to 400 m in the east.

The average altitude of the experimental area is 1219 m, with the highest point reaching 3955 m. The safe flight altitude is 600 m above the peak, resulting in a designed flight altitude of approximately 4600 m. Considering the designed flight altitude in relation to the average altitude, the mean aircraft height above the ground is determined to be 3381 m. Given an equipment incident angle of 47° with a variation range of 15°, the actual incident angle spans from 39.5° to 54.5°. Using Formula (5), the theoretical strip width is derived as follows:W_strip_ = 4600 m × (tan54.5° − tan39.5°) = 2830.29 m

Taking the Cessna aircraft as an example, with a flight speed of 70 m/s during data acquisition and a turning slope of 25°, and using Formula (6), the minimum turning radius can be considered to be as follows:70 × 70/(9.18 × tan25°) = 1144.67 m

With a turning radius of 1144.67 m, the aircraft requires approximately 2289 m to complete a 180-degree turn, which is nearly equivalent to the theoretical strip width. This implies that, in theory, the flight route can be executed in a sequential manner.

Assuming the radar equipment uses right-looking observation, and using 0° to 360° with a 1° interval to calculate the flight heading with the least terrain influence, Figure 15 shows the sum of the proportions of medium and high terrain influence areas and the overlay shadow area. It can be seen that the heading angle around 315° (right-looking) is relatively the least affected by terrain.

Considering that the survey area is nearly 30% percent flat area, which needs fewer flight paths, if we set the heading angle to 315°, the data acquisition along a single flight path will include part of the mountainous area to the north and the flat area to the south. To ensure adequate coverage of the mountainous area, the flight paths will need to be denser, which will increase flight costs. Thus, the flight path design is carried out along the long side, with a heading angle of 46.7°. The proportion of the area affected by terrain in this direction is 2.1%, which is not the maximum value. According to a 30% overlap rate, the interval between the flight paths is 1366.4 m. Therefore, the flight path interval is designed to be 2830.29 m; the flight path and strip coverage simulation results are shown in Figure 16b. The strip width is insufficient in the mountainous area.

Due to the influence of mountainous terrain, there are gaps between the strip coverage, which cannot fully cover the measurement area. Therefore, the interval between routes is reduced in the mountainous area; Figure 17 shows the designed flight paths and the simulation calculation results of strip coverage and overlay shadow distribution. The flight mission is executed following the flight path depicted in Figure 17a.

In areas that are easily accessible by road, a total of 102 control points were established: 18 of these were synchronized using corner reflectors, while the remaining 84 control points were measured post-flight. The distribution of ground control points is shown in Figure 18a, while the elevation inversion results are shown in Figure 18b. The overlapping areas between the strips using different types of control points can match each other very well. The results show that both synchronized and post-flight control points can provide proper control information for elevation inversion.

In the working area, four ground check point measurement sample areas were selected, with the spatial distribution shown in Figure 19a. Ground check points were chosen in areas that were not obstructed and where there were no abrupt changes in height. The elevation data of the corresponding points were measured as ground inspection data and compared with the elevation values obtained from the DEM at those locations. The results are shown in Table 4. It can be seen that among the 20 selected points, the difference between the inverted elevation and the measured elevation is less than 2 m, indicating that the digital terrain accuracy is better than a scale of 1:2000.

## 5. Discussion

The present study addresses the challenges of acquiring high-precision terrain data in complex mountainous regions, where traditional methods are often hindered by cloud cover, variable terrain, and restricted airspace. By employing an airborne millimeter-wave InSAR system, we have demonstrated a significant advancement in the efficiency and accuracy of terrain mapping under sparse synchronous control conditions.

Our approach, which integrates advanced route planning with global SRTM DEM data and radar observation geometry, has resulted in an optimal flight path design that minimizes the impact of terrain variations on data acquisition quality. This method not only streamlines the data collection process but also reduces the financial and logistical burdens associated with traditional field work and aviation operations.

The innovative reduction in synchronous ground control points, coupled with post-flight control measurements, has proven to be a viable strategy for enhancing the accuracy of elevation inversion. This strategy significantly decreases the reliance on the labor-intensive and error-prone field deployment of corner reflectors, thus improving the overall reliability of the data.

The experimental results from a selected area in the southwestern mountainous region of Sichuan have validated our methodology. The data acquired demonstrated a high degree of accuracy and consistency, aligning well with the expected outcomes of our simulation and route planning. The successful integration of synchronous and post-flight control points for elevation inversion has reinforced the robustness of our approach.

In conclusion, the airborne millimeter-wave InSAR system, supported by our optimized flight path design and control point strategy, offers a powerful tool for terrain mapping in complex mountainous areas. This research contributes to the field by not only improving the efficiency of data acquisition but also enhancing the accuracy of the resulting terrain models. The technical approach presented in the paper can be applied in regions that are similarly severely affected by cloud and fog meteorological conditions. It can be observed that the current technical procedures still require significant ground control measurement work. Furthermore, ground work will become more challenging in mountainous areas with limited road access. The goal of reducing reliance on ground control points remains a key direction for future technological advancements.

## Figures and Tables

**Figure 1 sensors-25-00424-f001:**
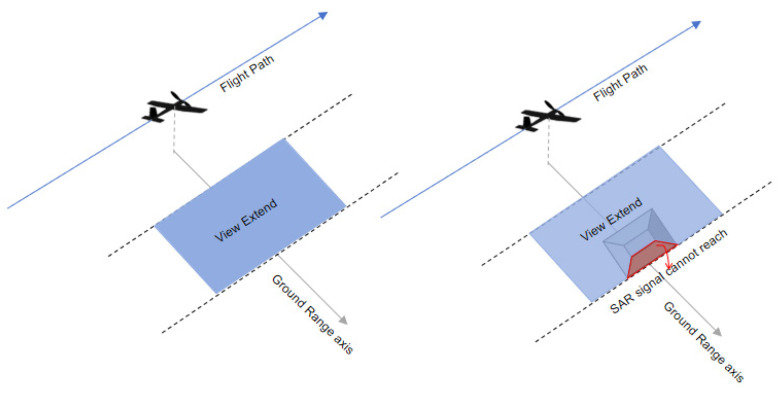
View extend diagram.

**Figure 2 sensors-25-00424-f002:**
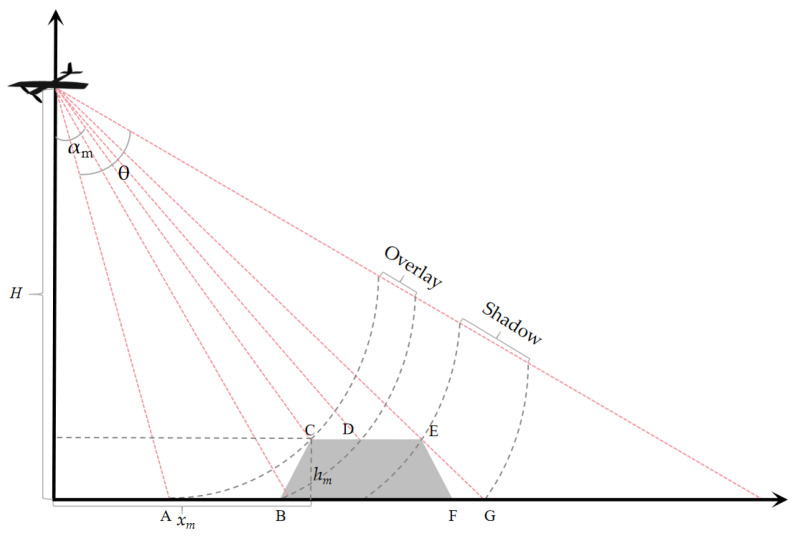
Shadow and layover schematic diagram, where grey area represents an object on the ground.

**Figure 3 sensors-25-00424-f003:**
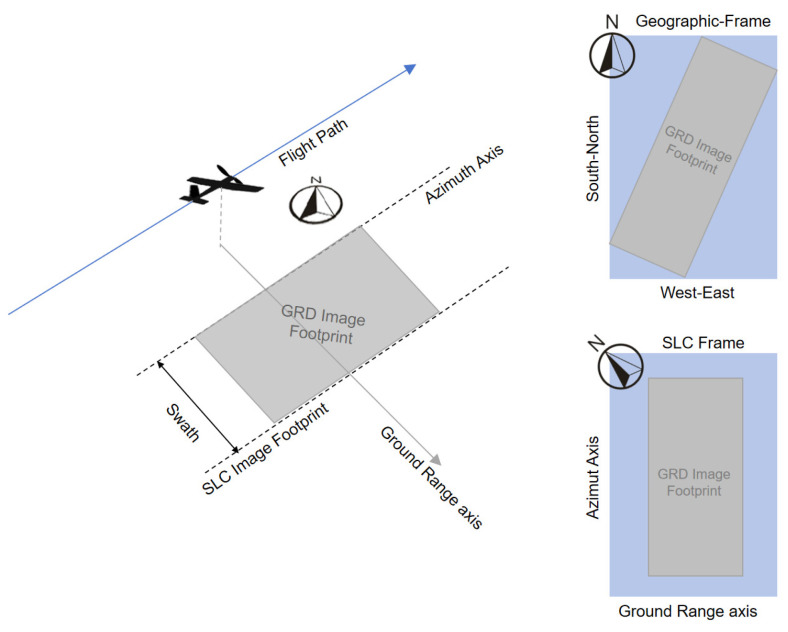
Convert DEM data to radar coordinate system.

**Figure 4 sensors-25-00424-f004:**
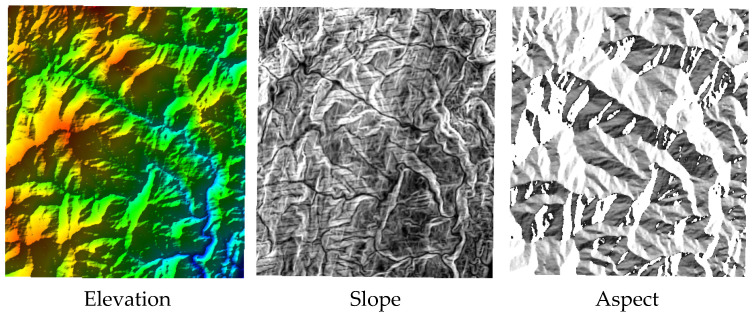
The main basic data for calculating R_Index.

**Figure 5 sensors-25-00424-f005:**
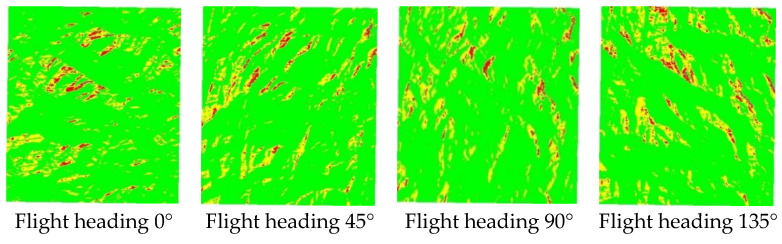
The evolution of different headings affected by terrain.

**Figure 6 sensors-25-00424-f006:**
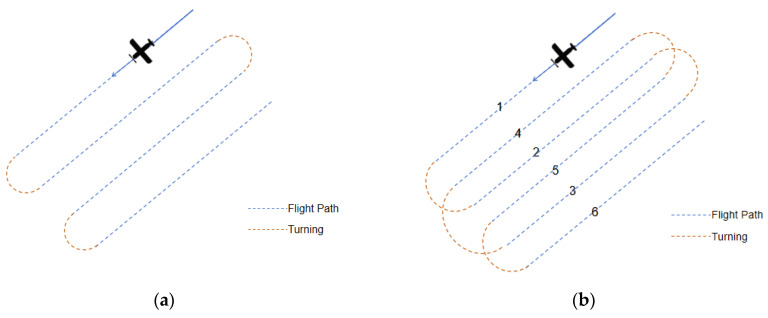
Flight path design, where (**a**) is the original flight path and (**b**) is the improved flight path of saving U-turn time by adjusting flight sequence from 1 to 6.

**Figure 7 sensors-25-00424-f007:**
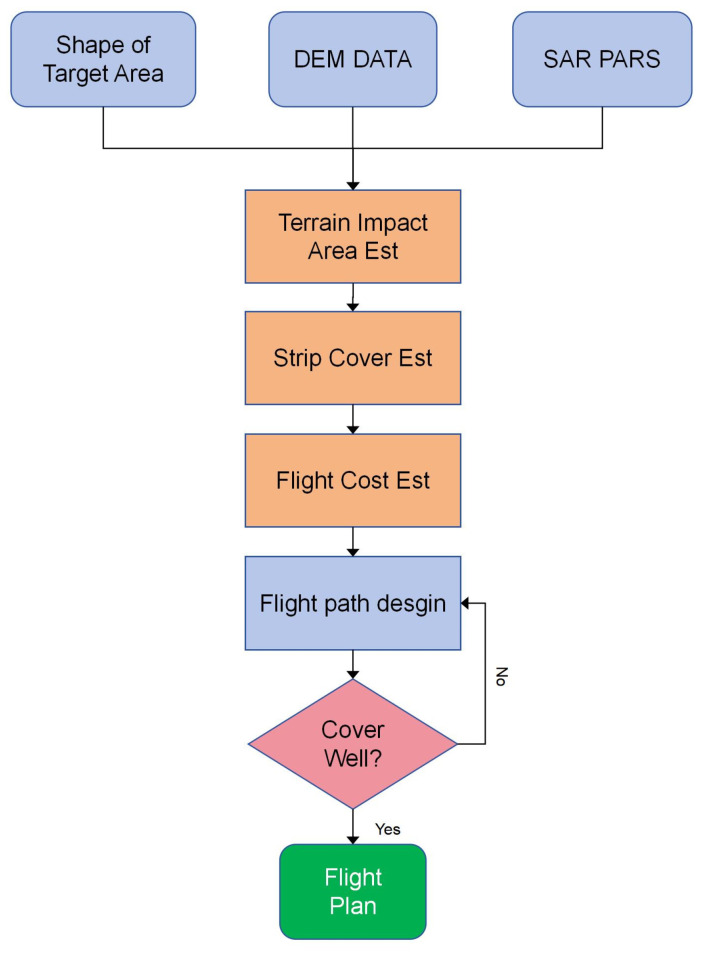
Work flow chart of flight path design.

**Figure 8 sensors-25-00424-f008:**
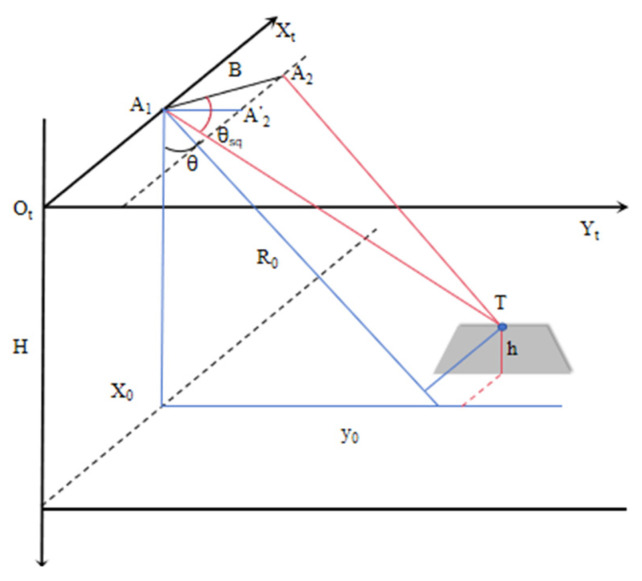
Radar antenna observation geometry diagram.

**Figure 9 sensors-25-00424-f009:**
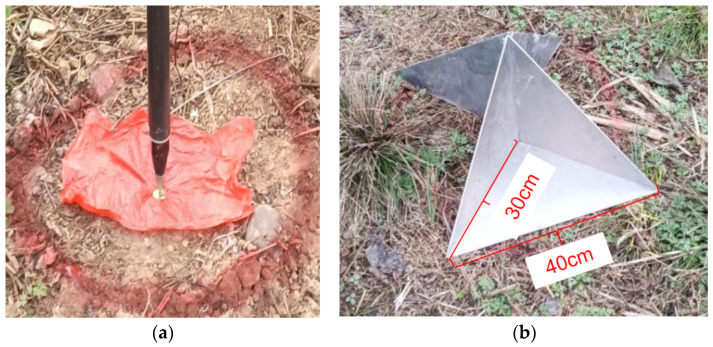
(**a**) Corner reflector placement position measurement and (**b**) corner reflector layout.

**Figure 10 sensors-25-00424-f010:**
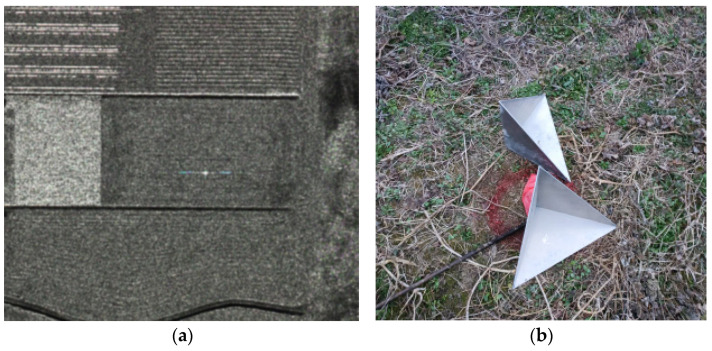
The imaging results of corner reflector diagram on radar image is shown in (**a**), and (**b**) is the corner reflector that was damaged and moved out of its original position.

**Figure 11 sensors-25-00424-f011:**
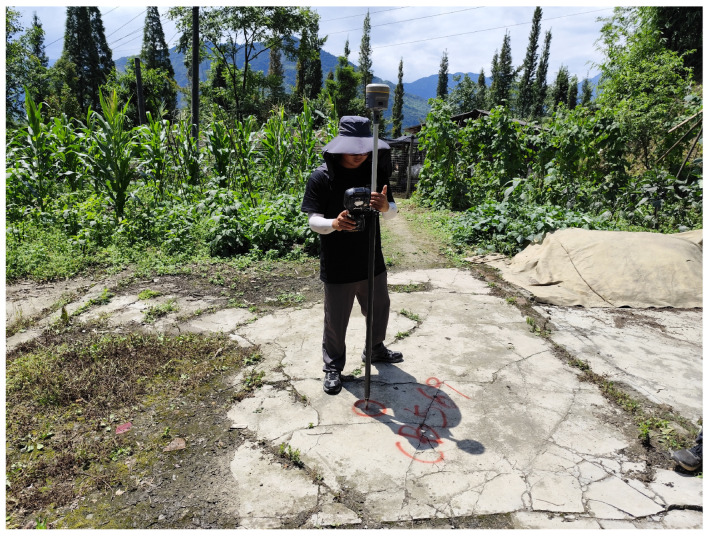
Supplementary control point measurements after the flight.

**Figure 12 sensors-25-00424-f012:**
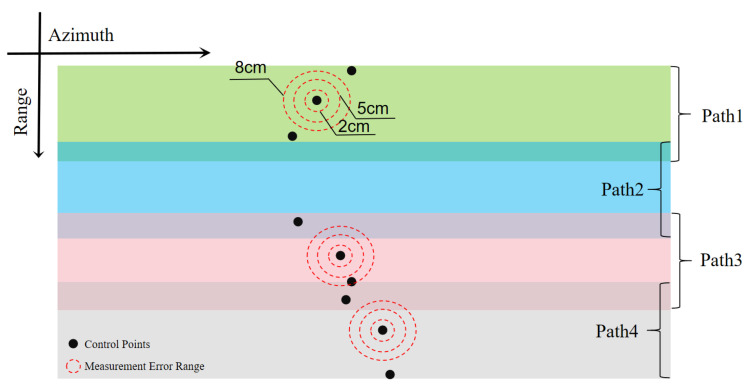
Measurement error experiment of manually adding control points.

**Figure 13 sensors-25-00424-f013:**
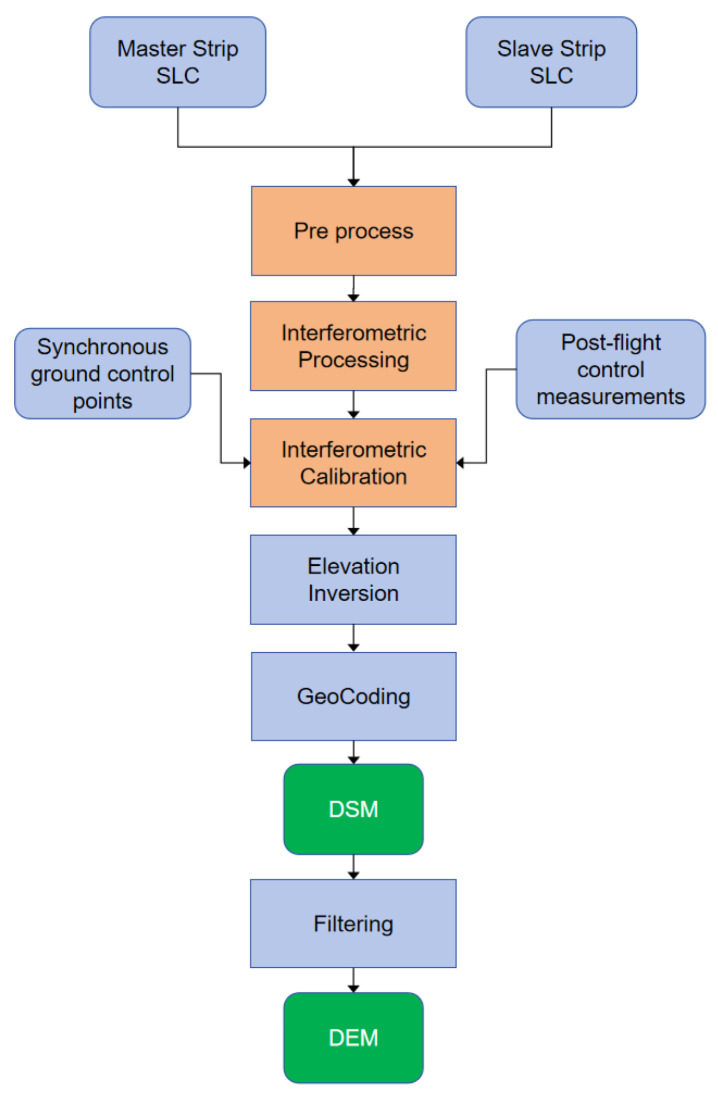
Airborne InSAR data processing flow chart.

**Figure 14 sensors-25-00424-f014:**
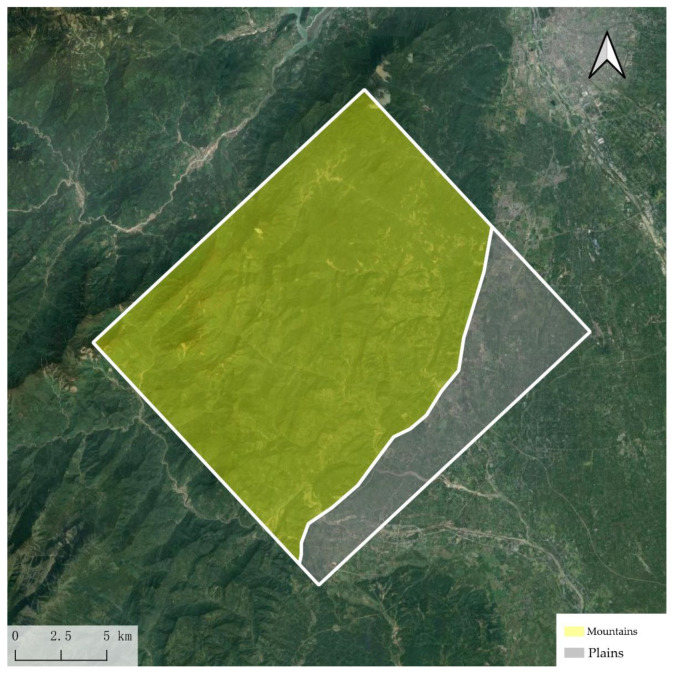
Working area.

**Figure 15 sensors-25-00424-f015:**
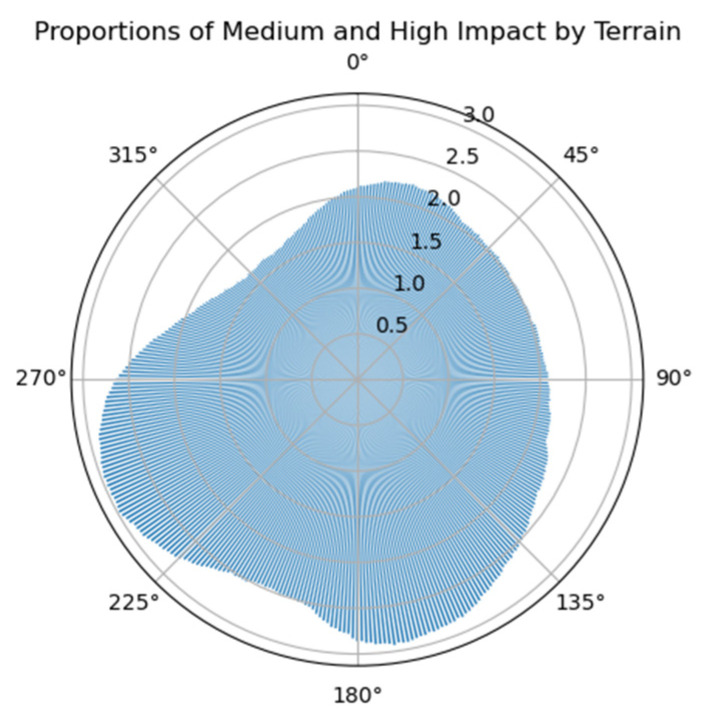
The relationship between the area proportion affected by terrain and flight heading angle.

**Figure 16 sensors-25-00424-f016:**
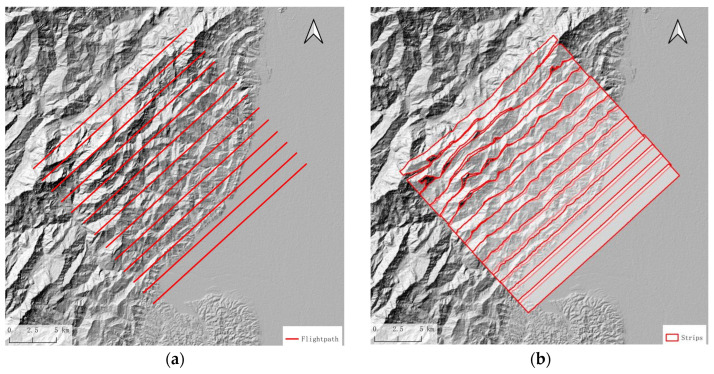
(**a**) Designed flight paths with the fix interval and (**b**) simulation results of equally spaced flight paths; there are gaps between each strip in the mountain area.

**Figure 17 sensors-25-00424-f017:**
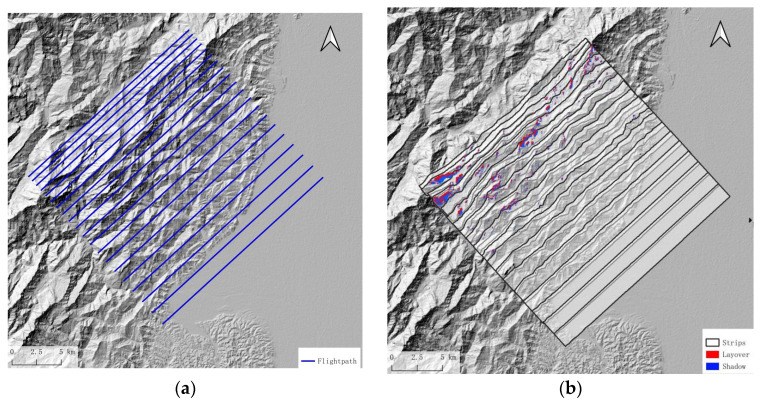
(**a**) Designed flight paths, which are denser in the mountain area, and (**b**) simulation calculation results of strip coverage with denser flight paths in the mountain area and overlay shadow distribution.

**Figure 18 sensors-25-00424-f018:**
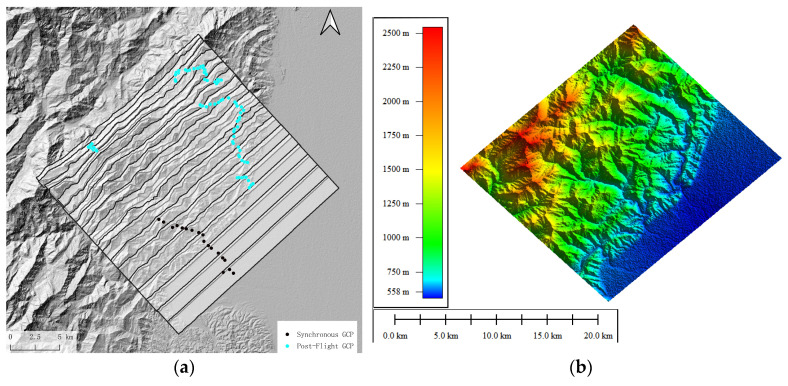
(**a**) Control point distribution. (**b**) Elevation inversion results.

**Figure 19 sensors-25-00424-f019:**
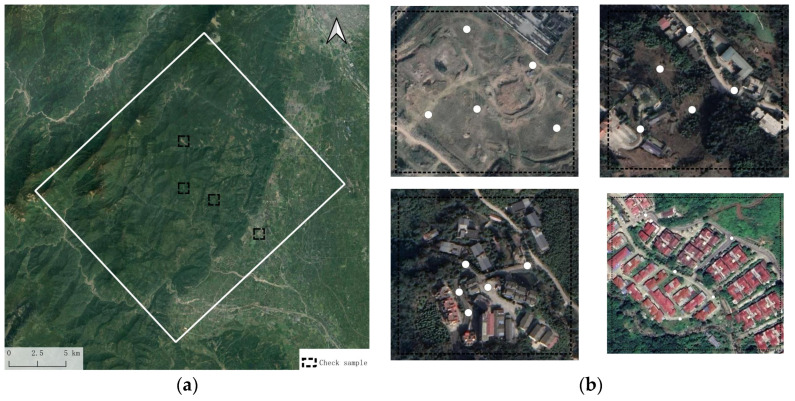
(**a**) Field control point measurement sample areas. (**b**) Distribution of ground check points.

**Table 1 sensors-25-00424-t001:** R_Index classification.

Value	Affected by Terrain	Color
Ri ≤ 0	Shadow/Layover	
0 < Ri ≤ 0.25	High	
0.25 < Ri ≤ 0.5	Medium	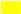
0.5 < Ri ≤ 1	Small	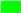

**Table 2 sensors-25-00424-t002:** The influence of control point measurement error on height measurement accuracy.

Control Point Measurement Error/cm	Mean Height Error/m	Standard Deviation of Height Error/m
0	−0.039	0.327
	−0.062	0.302
	−0.296	0.450
2	−0.019	0.326
	−0.042	0.302
	−0.276	0.449
5	0.011	0.324
	−0.013	0.301
	−0.246	0.449
8	0.041	0.322
	0.017	0.301
	−0.215	0.449

**Table 3 sensors-25-00424-t003:** Radar parameters.

Parameters	Value
Band	Ka
Incident angle (deg)	47
Radar beamwidth in elevation (deg)	15
Average flight speed (m/s)	72

**Table 4 sensors-25-00424-t004:** Comparison between DEM elevation and ground measurement results.

DEM Height (m)	Ground Measurement (m)	Height Change (m)	DEM Height (m)	Ground Measurement (m)	Height Change (m)
570.73	570.51	0.21	953.85	953.53	0.31
570.55	570.68	−0.13	952.24	952.79	−0.55
571.38	571.09	0.28	934.62	934.32	0.29
571.17	571.87	−0.70	947.46	947.06	0.39
570.81	570.32	0.48	917.18	917.78	−0.60
826.66	826.16	0.49	905.89	904.80	1.09
828.44	828.85	−0.41	902.33	902.59	−0.26
828.29	828.47	−0.18	928.76	928.02	0.73
842.52	842.25	0.26	904.61	903.99	0.62
859.71	859.04	0.67	943.1	943.68	−0.58

## Data Availability

Data is contained within the article.

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
