# Peer review of "Optimized Airborne Millimeter-Wave InSAR for Complex Mountain Terrain Mapping"

_sensors, 2025, doi:10.3390/s25020424_

Round 1
Reviewer 1 Report
Comments and Suggestions for Authors
1. Lines 122-123: The syntax for "object and sensor below point" is
incorrect and needs to be reorganized.
2, Lines 366-367: "18 synchronized control points were established using
corner reflectors, in addition to 84 control points measured after the
flight" the structure is not clear. This needs to be rephrased for clarity.
3. In the accuracy verification of the experimental area, the maximum
difference between the DEM results and the ground elevation is within 2
m. Is this level of accuracy sufficient to meet terrain mapping
requirements at 1:2000 scale? How do the experimental results support
the feasibility of the method?
Author Response
Comments 1: Lines 122-123: The syntax for "object and sensor below point" is incorrect and needs to be reorganized.
Response 1:Thank you for pointing this out. The sentence is changed into "the ground distance between the object and the sensor below"
Comments 2: Lines 366-367: "18 synchronized control points were established using corner reflectors, in addition to 84 control points measured after the flight" the structure is not clear. This needs to be rephrased for clarity.
Response 2: Thanks for your notice. This sentence has been changed into "A total of 102 control points were established: 18 of these were synchronized using corner reflectors, while the remaining 84 control points were measured post-flight."
Comments 3: In the accuracy verification of the experimental area,the maximum difference between the DEM results and the ground elevation is within 2 m. Is this level of accuracy sufficient to meet terrain mapping requirements at 1:2000 scale? How do the experimental results support the feasibility of the method?
Response 3:Thank you for pointing this out. According to "People's Republic of China Surveying and Mapping Industry Standards Digital products of fundamental geographic information 1:500 1:1000 1:2000 digital elevation models CH/T9008.2-2010". The elevation mean error of 1:2000 DEM data should be better than 2.25m in mountain areas. As a result the test result meet the 1:2000 height accuracy requirement.
Reviewer 2 Report
Comments and Suggestions for Authors
This paper investigates the use of airborne millimeter-wave InSAR systems for terrain mapping. The experimental results in the mountainous regions of Sichuan Province have validated the proposed method. The specific suggestions are as follows,
1. The numbering order of the main sections of the article needs to be carefully checked; the section number for Discussion should be 5.
2. Figure 15 should clearly indicate the levels affected by topography. In the previous text, the levels affected by topography were divided into four grades. It should be specified in the figure's caption which grades are affected, or it should be explained that the figure shows the sum of several affected grades.
3. Check the capitalization of the English initials in the figure legend and the background style of the legend, as there is an inconsistency in the capitalization of the initials in the legend of Figure 16.
4. The conclusion should summarize the key findings and their significance, and suggest potential areas for future research.
Author Response
Comments 1: The numbering order of the main sections of the article needs to be carefully checked; the section number for Discussion should be 5.
Response 1:Thanks for your notice. The numbering order has been checked and changed accordingly.
Comments 2: Figure 15 should clearly indicate the levels affected by topography. In the previous text, the levels affected by topography were divided into four grades. It should be specified in the figure's caption which grades are affected, or it should be explained that the figure shows the sum of several affected grades.
Response 2: Thanks for your notice.The title of Figure15 has been updated.
Comments 3: Check the capitalization of the English initials in the figure legend and the background style of the legend, as there is an inconsistency in the capitalization of the initials in the legend of Figure 16.
Response3:Thanks for your notice. The figure has been updated, and all figures has been checked and updated accordingly.
Comments 4: The conclusion should summarize the key findings and their significance, and suggest potential areas for future research.
Response 4: Thanks for your notice. The Discussion section has been updated accordingly in Lines 424-430.
Reviewer 3 Report
Comments and Suggestions for Authors
The paper presents the application of INSAR technology to measure surface DEM in the areas with complicated topography. The paper contains the description of the observation geometry and impact of the layovers and shadows on the efficiency of the mapping surface coverage. The need of the field calibration activities is mentioned and discussed in a large extent. From technological point of view the paper will be of interest to readers.
Some small remarks are:
Lines 51-52: “Airborne dual-antenna millimeter-wave InSAR, equipped with two antennas, one for emitting electromagnetic waves and the other for receiving ground echo signals simultaneously.”
Remark: This is a single-pass interferometric system, where one antenna is emitting signal, and the echo signal is acquired by two antennas simultaneously.
Eq. 7: under the square root there should be R^2*sin^2(teta_sq)...)
Lines 223-224: Authors state that: “Affected by factors such as temperature, humidity, and radar power-on time, the above 224 error parameters need to be re-calibrated in different flight missions [19-20].”
Remark: First of all, in the single-pass interferometry, the temperature and humidity do not affect the interferometric measurements. Well, slant range calibration measurements, the baseline length and tilt calibration are of importance really. And refs 19-20 do not discuss the temperature and humidity corrupting effects and respective calibration activities to remove temperature and humidity influence.
Table 3: The antenna beamwidth is in azimuth or in elevation?
In Fig 18, which points are synchronized control points, and which points are the control points measured after the flight?
Lines 414-415: authors state that “airborne millimeter-wave InSAR system, supported by our optimized flight path design and control point strategy, offers a powerful tool for terrain mapping in complex mountainous areas”. Their strategy proposed involves selection and inclusion of the post-flight control points. It seems to be attractive as it allows the decrease of manpower when establishing the network of control triangles. But there is a question about the accuracy of phase measurements of such a natural targets with reduced backscatter compared with real corner reflectors.
Author Response
Comments 1:Lines 51-52: “Airborne dual-antenna millimeter-wave InSAR, equipped with two antennas, one for emitting electromagnetic waves and the other for receiving ground echo signals simultaneously.”
Remark: This is a single-pass interferometric system, where one antenna is emitting signal, and the echo signal is acquired by two antennas simultaneously. Eq. 7: under the square root there should be R^2*sin^2(teta_sq)...)
Response 1: Thank you for the careful review and kindly reminder, this is a clerical error and has been corrected in the manuscript (Eq. 7).
Comments 2: Lines 223-224: Authors state that: “Affected by factors such as temperature, humidity, and radar power-on time, the above 224 error parameters need to be re-calibrated in different flight missions [19-20].”
Remark: First of all, in the single-pass interferometry, the temperature and humidity do not affect the interferometric measurements. Well, slant range calibration measurements, the baseline length and tilt calibration are of importance really. And refs 19-20 do not discuss the temperature and humidity corrupting effects and respective calibration activities to remove temperature and humidity influence.
Response 2: Dear reviewer, I'm so sorry that this is an obvious mistake. Temperature, humidity, and radar power-on time affect the initial phase of the system, but they do not affect the interferometric measurements. The main point in this section is that traditional interferometric calibration uses the same calibration field to calibrate these error parameters, and does not take into account the variations of parameters between different sorties and different strips. In fact, each strip needs to complete the interferometric calibration in order to obtain accurate elevation data. Revisions have been made to the manuscript and the corresponding references have been removed. (L227~L232).
Comments3: Table 3: The antenna beamwidth is in azimuth or in elevation?
Response 3:Thanks for your notice.The antenna beamwidth is in elevation, which has been explicitly described in the manuscript. (Table 3)
Comments4: In Fig 18, which points are synchronized control points, and which points are the control points measured after the flight?
Response 4: The dark dots in Figure 18 are synchronized control points(synchronized GCP), and the bright blue dots are the control points measured after the flight(Post-Flight GCP).
Comments5: Lines 414-415: authors state that “airborne millimeter-wave InSAR system, supported by our optimized flight path design and control point strategy, offers a powerful tool for terrain mapping in complex mountainous areas”. Their strategy proposed involves selection and inclusion of the post-flight control points. It seems to be attractive as it allows the decrease of manpower when establishing the network of control triangles. But there is a question about the accuracy of phase measurements of such a natural targets with reduced backscatter compared with real corner reflectors.
Response 5: Corner reflectors have strong back scattering, which allows them to be easily identified from SAR imaging data. Subsequently, the relationship between real XYZ coordinates and SAR coordinates can be established, thereby finally connecting the phase and ground height.
Post-flight control points, which serve as natural targets, exhibit reduced back scattering and are not easily identifiable from SAR imaging data. However, with the integration of GPS/IMU equipment into the synthetic aperture radar system, SAR data can be accurately geocoded to their correct geographical locations. This allows for the establishment of a connection between the measured height of specific ground point and the corresponding phase information. Nevertheless, it cannot be guaranteed that the GPS/IMU equipment will always function optimally; therefore, currently the deployment of corner reflectors remains necessary.
In out test mission measurements of post-flight control points are conducted within a few days following the completion of flight operations. As depicted in Figure 11, areas characterized by minimal ground obstruction and gradual topography are selected for post-flight height measurement targets. In mountainous regions with limited construction activity, it is relatively simple to identify flat roadside locations that remain stable over extended periods and exhibit good coherence. Therefore, we use these Post-Flight control points for as supplementary for synchronized Corner reflector GCPs.
Round 2
Reviewer 2 Report
Comments and Suggestions for Authors
Thank you to the author for the detailed revisions and responses. I have no further concerns.